# Ruminococcaceae_UCG-013 Promotes Obesity Resistance in Mice

**DOI:** 10.3390/biomedicines10123272

**Published:** 2022-12-16

**Authors:** Jinlian Feng, Hongliang Ma, Yiting Huang, Jiangchao Li, Weidong Li

**Affiliations:** 1School of Nursing, Guangdong Pharmaceutical University, Guangzhou 510006, China; 2School of Life Sciences and Biopharmaceuticals, Guangdong Pharmaceutical University, Guangzhou 510006, China; 3School of Health, Guangdong Pharmaceutical University, Guangzhou 510006, China

**Keywords:** obese prone, obese resistant, 16S rRNA gene sequencing, intestinal flora, Ruminococcaceae_UCG-013

## Abstract

Alterations in the gut microbiome have been linked to obesity and type 2 diabetes, in epidemiologic studies and studies of fecal transfer effects in germ-free mice. Here, we aimed to identify the effects of specific gut microbes on the phenotype of mice fed a high-fat diet (HFD). After eight weeks of HFD feeding, male *C57BL/6J* mice in the HFD group ranking in the upper and lower quartiles for body weight gain were considered obese prone and obese resistant, respectively. 16S rRNA gene sequencing was used to determine the composition of the intestinal microbiota, and fecal transplantation (FMT) was conducted to determine whether the microbiota plays a causal role in phenotypic variation. Ruminococcaceae_UCG-013 was more abundant in the gut microbes of mice with a lean phenotype than in those with an obese phenotype. Ruminococcaceae_UCG-013 was identified as the most significant biomarker for alleviating obesity by random forest analysis. In a correlation analysis of serum parameters and body weight, Ruminococcaceae_UCG-013 was positively associated with serum HDL-C levels and negatively associated with serum TC, TG, and LDL-C levels. To conclude, Ruminococcaceae_UCG-013 was identified as a novel microbiome biomarker for obesity resistance, which may serve as a basis for understanding the critical gut microbes responsible for obesity resistance. Ruminococcaceae_UCG-013 may serve as a target for microbiome-based diagnoses and treatments in the future.

## 1. Introduction

Complex gene–environment interactions are considered necessary in the development of obesity [1]. The gut microbiota composition can determine the conversion efficiency of food energy [2]. Obesity increases the risk of many chronic diseases, such as type 2 diabetes, heart disease, and some cancers, and reduces life expectancy. Therefore, this issue must be addressed and prevented as soon as possible [3]. Aside from monogenic obesity, common obesity is a multifactorial condition involving genetic predispositions and environmental factors. A dietary intervention to reduce weight is considered a relevant treatment strategy for obesity since diet is a major environmental component [4].

Nevertheless, clinical trials examining the relationship between total caloric intake and body weight have reported highly variable and inconsistent results [5,6,7,8,9]. Microbes inhabiting the gut are gaining recognition as metabolic organisms that play a role in the metabolic network of the host. This impacts an individual’s physiology by influencing metabolism, immunity, ageing, and behaviour [10,11]. The gut microbiota has recently been identified as a regulator of energy metabolism that mediates the host’s response to dietary changes [12,13]. A study performed on germ-free mice revealed that obesity can be transmitted via faecal microbiota transplantation, suggesting the relevance of the gut microbiota to the deposition of adipose tissue. In a healthy gut microbiota, multiple commensal microbiotas play various roles. The microbiota in these environments may play an essential role in regulating energy balance by fermenting dietary fibres and bioconverting dietary components into metabolically bioactive molecules. In metabolic diseases, as well as in the metabolism of the host, the gut microbiota plays an important role. Numerous studies have consistently demonstrated that the gut microbiota influences the phenotypes of its hosts, whether they are obese or lean [14,15,16,17]. When examining the role of the gut microbiota in obesity, the abundance of phylum Bacteroidetes was lower in obese mice and humans, and the abundance of phylum Firmicutes was significantly higher than that in lean individuals [14,18]. However, it is unclear which specific bacterial species are involved in obesity resistance.

Metabolic capacity can be impacted by decreased species richness and diversity of the gut microbiota in obese humans and animals [2,19,20]. It is unclear precisely which gut flora contributes to obesity resistance in high-fat diet (HFD)-induced obesity. HFD-induced changes in the components and function of the gut microbiota in obese-prone (OP) and obese-resistant (OR) phenotypes are of interest to us. Using 16S rRNA gene sequencing, the intestinal microbiota was investigated. The Phylogenetic Investigation of Communities by Reconstruction of Unobserved States (PICRUSt) is a technique for predicting functional genes in metagenomics that was implemented to predict microbial metabolic functions. Additionally, the correlation between intestinal microbes and lipid metabolic parameters was examined using correlation analysis and visualized through a network. Our study aimed to identify specific gut microbiota constituents as potential targets for weight loss.

## 2. Materials and Methods

### 2.1. Animals and Experimental Design

Six-week-old *C57BL/6J* male mice were purchased from Sipeifu Biotechnology Co., Ltd. (Beijing, China) and maintained under specific pathogen-free (SPF) conditions. The animal ethics committee at Guangdong Pharmaceutical University approved our study. A strict adherence to the Guide for the Care and Use of Laboratory Animals developed by the National Institutes of Health was observed during all experiments.

Mice were fed a standard chow diet during an initial acclimatization period of 1 week. Following the random selection of mice for the control group, six mice were provided a steady standard chow diet consisting of 10% fat, 71% carbohydrates, and 19% protein as a standard diet. Meanwhile, 54 mice were fed an HFD containing 60% fat and 20% protein. Appendix A presents the detailed composition and percentage of the HFD (Nantong Trophy Experimental Animal Feed Co., Ltd., Nantong, China). Body weights and food intake were recorded weekly during the experiments. Based on a previous publication, mice in the HFD group ranked in the upper and lower quartiles for body weight gain were characterized as obesity-prone (OP, *n* = 12) and obesity-resistant (OR, *n* = 12), respectively [20,21]. The remaining mice were used in other experiments. Fresh faeces were collected from six mice from each of the three groups after eight weeks of feeding. These faeces samples were divided into sterile EP tubes with 10 μL of 1% sodium azide (NaN_3_) and stored at −80 °C for subsequent 16S rRNA gene sequencing analysis.

In the fecal transplantation (FMT) experiment, 24 male mice were randomly divided into three groups and fed an HFD consisting of 60% fat, 20% carbohydrates, and 20% protein. In this study, the weekly weight and food intake were recorded. As soon as the animals were adjusted to their living conditions, they were randomly divided into three groups: control (*n* = 8), Mice with obesity-prone fecal flora transplanted (FMT-OP) (*n* = 8), and Mice with obesity-resistant fecal flora transplanted (FMT-OR) (*n* = 8). For 14 days, mice in each group were given an ad libitum cocktail of antibiotics in drinking water to deplete the microbiota in the gut. There were four antibiotics in the antibiotic cocktail: ampicillin (1 g/L), metronidazole (1 g/L), neomycin (1 g/L), and vancomycin (500 mg/L). Fourteen days after the animals were given antibiotics, they underwent a washout period of seventy-two hours. Then, a total of 12 faecal samples were collected from each of the OP and OR groups after eight weeks of HFD feeding. Each 100 mg of faeces was resuspended in 1 mL of PBS and homogenized. Then, after centrifugation at 800 rpm for 5 min, the supernatant was harvested as an inoculum. A daily inoculum was prepared and immediately administered to the mice orally by gavage in a 200-micro-litre volume to minimize changes in the microbiota composition. In the control group, mice were given a gavage of PBS as a control for grafting.

Following the oral glucose tolerance test (OGTT), all mice were sacrificed immediately. Samples of liver tissues, intestinal tissues, and fat tissues were collected immediately following euthanasia. A liver sample was fixed in 4% paraformaldehyde, processed routinely, and embedded in paraffin. Paraffin-embedded liver sections were prepared and stained with haematoxylin and eosin (H&E). Other samples were stored at −80 °C until used for analysis.

### 2.2. Oral Glucose Tolerance Test

All mice were tested for oral glucose tolerance at the end of the experiment. The mice were fasted for eight hours before administering a glucose solution (2 g/kg) by oral gavage. An Ascensia Elite XL glucometer (Bayer AG, Zurich, Switzerland) was used to measure blood glucose levels at 0, 15, 30, 60, and 120 min.

### 2.3. Serum Biochemical Analysis

A 1.5 mL centrifuge tube was used to collect blood samples from the heart, and the blood sample was placed at room temperature for one hour. After centrifuging the serum at 3000× *g* for 15 min, it was stored at −80 °C. A commercial kit purchased from Nanjing Jiancheng Bioengineering Institute Co., Ltd. (Nanjing, China) was used to determine the serum concentrations of total cholesterol (TC), triglycerides (TGs), low-density lipoprotein-cholesterol (LDL-C), and high-density lipoprotein-cholesterol (HDL-C).

### 2.4. High-Throughput Sequencing Analysis of the Intestinal Microbiome

Bacterial DNA was extracted from faecal samples with a QIAamp Fast DNA Stool Mini Kit (Qiagen, Cat# 51604). PCR was conducted with barcoded specific bacterial primers targeting the variable regions 3–4 (V3–V4) of the 16S rRNA gene: forward primer 338F: 5′-ACTCCTACGGGAGGCAGCA-3′ and reverse primer 806R: 5′-GGACTACHVGGGTWTCTAAT-3′ [22]. Sequencing libraries and paired-end sequencing were constructed according to standard protocols using an Illumina NovaSeq6000 platform. Paired-end reads were merged using FLASH (version 1.2.11) [23] and tags with more than six mismatches were discarded. Merged tags with a mean quality score of less than 20 within a 50-bp sliding window were identified using Trimmomatic [24], and tags shorter than 350 bp in length were removed. Potential chimaeras were further removed, and the denoised sequences were grouped into operational taxonomic units (OTUs) with 97% similarity using the program USEARCH (version 11.0, 2018.7). All OTUs were ranked by searching the Silva database (Release 128) using the software package QIIME (versoin 2020.6). The original sequences were stored in the Sequence Reading Archive (http://www.ncbinlm.nih.gov/sra (accessed on 23 November 2022) database with logarithmic numbers ranging from SAMN 32065991 to SAMN 32066008.

### 2.5. Quantitative PCR Analyses

Gene DNA was extracted from faecal samples and caecal contents using a QIAamp DNA Stool Mini Kit (Qiagen, Valencia, CA, USA) with a modified method, including a bead-beating procedure. Each sample was mixed with 0.2 g zirconia/silica beads (diameter: 0.1 mm, Biospec, GER) and lysis buffer. RNase was used to remove RNA. A concentration column was run before elution of the eluate. Single DNA was stored at −20°C. For quantitative RT–PCR, 1× SYBR Green Master Mix Buffer (Takara, Otsu, Japan) was used, and assays were run on a LightCycler 96 PCR apparatus. Common primers were 338F (5′-CCTACGGGAGGCAGCAG-3′) and 806R (5′-GGACTACHVGGGTWTCTAAT-3′). The primer sequences for Ruminococcaceae_UCG-013 are shown in Appendix A

### 2.6. Statistical Analysis

Data are presented as the mean ± SD of at least two independent experiments. Significant differences between groups were analysed by one-way analysis of variance (ANOVA) at *p* < 0.05 using SPSS (IBM, Armonk, NY, USA) and Prism 9 (GraphPad, San Diego, CA, USA).

## 3. Results

### 3.1. Phenotypes and Serum Parameters of Different Groups of Mice

After 16 weeks of feeding an HFD or chow diet, the phenotypic characteristics, body weight, and organ weight were examined. Representative images of mice are shown in Figure 1A. It was noted that mice in the HFD group responded differently to the HFD. Based on the changes in body weight between the Normal diet (ND), obesity-prone (OP), and obesity-resistant (OR) mice during the feeding period (Figure 1B), body weight gain was highest in the OP mice, followed by the ND and OR mice. According to the results of the study, the final body weight of the OP mice was 68% heavier than that of the OR and ND mice. There was no difference in the average energy intake between the two groups of mice on the HFD (Figure 1C). In comparison with the OR and ND mice, the OP mice had higher subcutaneous, epididymal, perirenal, and brown fat masses (Figure 1D). Furthermore, the OP mice had higher fat ratios after normalizing for body weight. At 0, 15, 30, 60, and 120 min during the OGTT, the blood glucose levels of the OP mice increased significantly compared to those of the OR and ND mice (Figure 1E). The OP mice had significantly higher serum TG, TC, and LDL-c levels than the ND and OR mice (Figure 1F). Haematoxylin and eosin staining (H&E) was used to determine liver lipid deposition and steatosis in each group (Figure 1G. Liver lipid deposition was absent in the ND mice, and the morphology of their tissues was typical. However, the HFD mice developed steatosis and hepatocyte ballooning in the liver and had higher levels of lipid droplets. Compared to the OR mice, the OP mice had more lipid deposition, inflammatory cell infiltration, and hepatocyte degeneration (Figure 1G). While both the OP and OR mice were fed the same obesogenic diet and maintained under the same conditions, they showed differing susceptibilities to obesity. Overall, the OR mice had a healthier overweight/obesity phenotype (HO status).

### 3.2. Gut Microbiota Differ between OP and OR Mice Fed the Same High-Fat Diet

There is increasing evidence that the gut microbiota plays a vital role in metabolic regulation. Consequently, we analysed the composition of the gut microbiota of the OP and OR mice. To map the microbial structure microbiota in the OP and OR subjects, 16S sequencing (V3–V4) and computational analysis were performed. By sequencing 16S rRNA genes using Illumina NovaSeq 6000, the microbial communities within faeces were determined. A total of 1,436,513 effective sequences were recovered with a read length of 360 base pairs. Following quality filtering by QIIME, 1,392,326 high-quality reads were obtained, accounting for 96.92% of the raw reads. Bacterial abundance was calculated using the ACE index and Chao1 index; bacterial diversity was calculated using the Shannon index and Simpson index. An analysis of the alpha diversity showed that the intestinal flora structure diversity (including the Chao index, Ace index, Shannon index, and Simpson index) was significantly lower in the HFD group than in the ND group under the same measurement depth. The Chao-1 and ACE indices in OP mice were considerably higher than those in OR mice, while the Shannon and Simpson indices showed no statistically significant differences (Figure 2A). Species diversity among samples was shown by partial least squares discriminant analysis (PLS-DA). The composition of microbial species was more similar when samples were closer to one another. According to the PLS-DA at the species level, the gut microbiota composition differed significantly among the three groups. There was an apparent separation between the OR and OP groups (Appendix A). A comparison was made between the relative abundances of bacterial taxa after mice were fed an HFD based on their phenotype (OP mice vs. OR mice). The Phylum Firmicutes was found to be more abundant in the OR group, whereas Bacteroidetes were more abundant in the OP group (Appendix A). More taxonomic groups were observed in the OR mice than in the controls, even in small amounts, as exemplified by class Melainabacteria (Appendix A). Clostridiales and Verrucomicrobiales were found to be more abundant in the OR group (Appendix A), and numerous studies have been reported to be implicated in diet-induced obesity and to correlate inversely with body weight [25,26]. The families Akkermansiaceae and Ruminococcaceae were more abundant in the OR group, whereas Leuconostocaceae and Burkholderiaceae were more abundant in the OP group (Appendix A). STAMP software analysis revealed that there were specific microbial phylotypes among the OP and OR groups based on taxonomic analysis (Figure 2C). Compared to the OP group, the OR group demonstrated significantly greater relative abundances of [Eubacterium]_coprostanoligenes_group, Akkermansia, and Ruminococcaceae_UCG-013; however, fewer Mucispirillum, Leuconostoc, [Ruminococcus]_torques_group, and GCA-900066225 were found in the OR group. Therefore, the gut microbial composition of the OP mice differed significantly from that of the OR mice.

### 3.3. Obesity Resistance-Associated Gut Microbiota Screening

Considering the apparent differences in the composition of the gut microbiota between the OR and OP groups, linear discriminant analysis effect size (LEfSe) analysis was used to investigate the microbiota enrichment over the study duration. Different taxa were abundant in different groups using LEfSe, which is an algorithm for discovering biomarkers in high dimensions using linear discriminant analysis (LDA) to identify solid statistical correlations among group members. According to the LEfSe analysis, the main biomarkers of the OP group were Deferribacteres, Brachyspira_sp, [Ruminococcus]_torques_group, and Leuconostocaceae, whereas the main biomarkers of the OR group were Ruminococcaceae_UCG-013, [Eubacterium]_coprostanoligenes_group, and Verrucomicrobia (Figure 3A). A random forest algorithm was applied to explore factors contributing to the observed microbial patterns. Ruminococcaceae_UCG-013 was identified as the signature microbe for distinguishing OP and OR mice (Figure 3B). Ruminococcaceae_UCG-013, a cellulose- and hemicellulose-degrading bacterium, is closely associated with metabolic diseases [27]. Moreover, Ruminococcaceae_UCG-013 was the least-abundant species among the OP groups at weeks 8 and 16 (Figure 3C,D). Based on a heatmap and network analysis (Figure 3E,F), the relative abundance of several critical intestinal microbial phylotypes was significantly correlated with serum parameters and body weight According to Figure 3F, Ruminococcaceae_UCG-013 (enriched in the OR group) exhibited a negative correlation with serum levels of TC, TGs, and LDL-C, and body weight. However, it showed a positive correlation with HDL-C levels. In conclusion, the most critical “biomarker” to distinguish OP from OR mice was Ruminococcaceae_UCG-013.

### 3.4. The PICRUSt Algorithm Predicts the Metabolic Function of Microbes

PICRUSt predicted the differences between the microbial metabolic functions of OP and OR mice based on the KEGG database (Figure 4). In the OR group, the phosphotransferase system (PTS), Alanine, aspartate and glutamate metabolism, and Folate biosynthesis were enriched. Folate was associated with a lower risk of weight gain in adults. Several studies have suggested that it is functionally related to the lean phenotype status. The OP group showed enriched levels of LPS biosynthesis pathways, protein digestion and absorption pathways, “Valine, leucine and isoleucine degradation”, and Arachidonic acid metabolism pathways suggesting that these pathways may play a significant role in obesity.

### 3.5. The Relationship between the Gut Microbiota and Obesity Phenotype

The caecal microbiota from OP and OR mice was colonized with antibiotic-treated *C57BL/6* mice to better understand the causal relationship between the microbiota and metabolic syndrome. Antibiotic-treated animals were started on an HFD one week before colonization and maintained on this diet for eight weeks after receiving faecal transplants from the OP and OR donors (*n* = 8 recipients/donor microbiota). Figure 5 shows the phenotypic characteristics, body weight, and organ weight after eight weeks of FMT. Different groups of intestinal flora were transplanted into mice in the FMT group, and their responses varied. Figure 5A presents representative images of mice. As shown in Figure 5B, the FMT-OP mice gained the most body weight, followed by the FMT-PBS and FMT-OR mice. In comparison with the FMT-OR mice, the FMT-OP mice had a higher subcutaneous, epididymal, and perirenal fat index (Figure 5D). The FMT-OR and FMT-OP mice consumed similar amounts of energy on average (Figure 5C). There were significant differences between the FMT-OR and FMT-OP mice in terms of serum TG, TC, HDL-c, and LDL-c levels (Figure 5F). At 0, 15, 30, 60, and 120 min during the OGTT, blood glucose levels were significantly higher in the FMT-OP mice than in the FMT-OR mice (Figure 5E). In addition, Ruminococcaceae_UCG-013 was the least abundant species in the FMT-OP group at week 8 of FMT (Figure 5G). The livers of the FMT-OP mice exhibited more significant levels of lipid deposition, inflammatory cell infiltration, and hepatocyte degeneration than those of the FMT-OR mice (Figure 5H). In conclusion, the above findings indicate that intestinal flora plays a critical role in obesity development.

## 4. Discussion

In the present study, we sought to identify specific gut microbes associated with the lean phenotype. Although the OP and FMT-OP mice consumed a similar amount of calories, they gained more weight and acquired more abdominal fat than the OR and FMT-OR mice. The metabolic phenotypes of the OR and FMT-OR mice were healthier than those of the OP and FMT-OP mice. Furthermore, 16S rRNA sequencing revealed that the gut microbiota of the OR mice differs significantly from those of the OP mice in terms of structure and function. As a result, we believe it is essential to investigate the specific bacteria that contribute to obesity resistance. In this study, we found that Ruminococcaceae_UCG-013 had a higher relative abundance in the gut microbiota of the lean phenotype than in those of the obese phenotype. Our findings suggest that Ruminococcaceae_UCG-013 may play an essential role in obesity resistance.

From the OGTT results, it can be concluded that the blood glucose levels of mice in the OR and FMT-OR groups were significantly lower than those of mice in the OP and FMT-OP groups. Obesity is traditionally believed to be a result of external influences, but there is evidence that one of the most important influences on obesity may originate inside the gut of an individual [28]. Recent studies of the human microbiome support the idea that the condition of one’s microbiome may favour obesity, inflammation, and insulin resistance, ultimately predisposing to T2DM [28]. This also explains why the OP group is hyperglycaemic in the same dietary setting. Further research into the relationship between specific gut flora and blood glucose is therefore essential to uncover potentially effective treatments for T2DM, including solutions that harness the potential of the gut microbiota.

Since the HFD-fed mice were fed identical diets, this study focused on the microbiota rather than diet exposure and obesity development. These findings suggest that ecological dysbiosis of the microbiota in the HFD feeding system is a factor in weight gain rather than a consequence of it. The intimate association between obesity and the gut microbiota is widely accepted [16,17,29]. The results of this study and previous studies [30,31,32] demonstrated that OR mice remained leaner than OP mice when fed an HFD and that the gut microbiota of OR and OP mice differed significantly. OR mice had a greater abundance of Ruminococcaceae. Ruminococcaceae (one of the four family-level taxa that discriminate lean from obese) is a major butyrate producer and a crucial bacterium for the health of the intestines [25,33,34,35]. In the HFD-fed mice, Ruminococcaceae_UCG-013 was identified as the most critical obesity resistance-related biomarker. In a correlation analysis of intestinal microbial phylotypes, serum parameters, and body weight, Ruminococcaceae_UCG-013 showed a positive correlation with HDL-C levels but a negative correlation with serum TC, TG, and LDL-C levels, and body weight. Thus, Ruminococcaceae_UCG-013 may play a significant role in preventing weight gain. Previous studies have reported that obesity is associated with a higher ratio of Firmicutes to Bacteroidetes [36], but our study demonstrates the opposite. However, other studies have reported contradictory results [37,38].

Ruminococcaceae_UCG-013 belongs to the Ruminococcaceae family, and many bacteria in this family are known to be general butyrate-producing bacteria. In two independent longitudinal Chinese cohort studies (a discovery cohort and a validation cohort), Ruminococcaceae_UCG_013 was enriched in populations with high dietary diversity, particularly in participants who consumed a high diversity of fruits and dairy products [39]. A significant negative correlation was found between Ruminococcaceae_UCG-013 and TG levels and a significant positive correlation was found between HDL-C and Ruminococcaceae_UCG-013 [40]. By interacting with barley, Ruminococcaceae_UCG-013 may ameliorate dyslipidemia in people with a high barley intake [41,42]. Based on a cross-sectional study on the relationship between the gut microbiota and childhood obesity, Ruminococcaceae_UCG_013 may play an important role in weight loss in children [43]. Among overweight and obese adults who lost between 5 and 10% of their body weight, Ruminococcaceae_UCG-013 was found to be enriched [44]. In summary, Ruminococcaceae_UCG-013 may play an important role in weight loss; however, the mechanism by which Ruminococcaceae_UCG_013 affects body weight needs further investigation.

We used PICRUSt to predict potential metagenomes based on the community profiles of 16S rRNA genes to investigate the relationship between obesity and gut microbiome functions. The OR group was enriched in folate biosynthesis, Alanine, aspartate and glutamate metabolism. Folate was linked to a lower risk of weight gain in adults and is functionally associated with lean phenotypes. Regarding polysaccharide (LPS) synthesis, arginine and proline metabolism, Protein digestion and absorption, Valine, leucine and isoleucine degradation, and Arachidonic acid metabolism, the microbiota from OP mice differed significantly from those of OR mice. LPS is known to be involved in the development of obesity as a direct target molecule for lipid delivery and storage within adipose tissue and plays an essential role in inflammation.

Adipose tissue infiltration by macrophages, CD8+T cells, and CD4+T cells is associated with obesity in mice and humans [45]. Several amino acids, such as cysteine, glutamine, phenylalanine, tryptophan, and arginine, have a regulatory effect on T-cell proliferation and activation as well as the subsequent immune response [46]. The results of a 16-week randomized controlled trial indicate that a methionine-restricted diet increases fat oxidation [47]. Sulfur amino acid (SAA) restricted diets reduced plasma methionine, cystathionine, and urinary total cysteine levels in adipose tissues, increased serum fibroblast growth factor 21 (FGF21) levels and altered the expression of gene products in subcutaneous adipose tissues [48]. A low SAA diet is likely to have beneficial effects on body composition and metabolic health, thus, opening up new strategies for preventing and treating overweight, obesity, and their associated diseases.

Based on macrogenomic studies of the human gut microbiota and microbiome, we know that early postnatal environmental exposures influence the overall phylogenetic structure of the human gut microbiota. During the initial stages of life, the human gut microbiota undergoes a process of ecological succession, which culminates in the establishment of a relatively stable and complex community. In the first three years of life, the microbiota is assembled into adult structures [49]. Human gut microbiomes are shared among family members, but the specific bacterial present in each person’s gut microbiome vary, and similar differences have been found in the microbial profile of twins [50,51]. The composition of the gut flora is driven by diet, antibiotic treatment, maternal microbiota, and genotype [52,53]. Microbial genomic variation includes Single Nucleotide Polymorphism (SNP), Short Insertion Deletion (Indel), and Structure Variation (SV). The early gut microbiome is influenced by many factors, including maternal diet, obesity, smoking status, mode of delivery, antibiotic use during pregnancy, antibiotic use in infants, and postnatal feeding patterns [54,55]. An individual’s gut microbiome in early life plays an important role in the development of their immune system and metabolism, which may influence their risk of chronic diseases (e.g., allergies, obesity, and other chronic immune and metabolic diseases in later life) [54].

A limitation of this study is that no single bacteria was tested. It is, therefore, unclear how Ruminococcaceae_UCG-013 affects obesity. Our subsequent study will examine the association between Ruminococcaceae_UCG-013 abundance in the intestinal microbiota and host health by isolating Ruminococcaceae_UCG-013 from intestinal microorganisms. To establish firm conclusions, we must emphasize that our sample size is small.

## 5. Conclusions

The relative abundance of Ruminococcaceae_UCG-013 in mice with a lean gut flora was consistently higher than that in mice with fat phenotypes fed the same diet. Correlation analysis revealed a negative correlation between Ruminococcaceae_UCG-013 and body weight. Ruminococcaceae_UCG-013 is one of the most critical cellulose- and hemicellulose-degrading bacteria and plays a crucial role in fibre degradation. As a result, leanness is associated with Ruminococcaceae_UCG-013. Furthermore, Ruminococcaceae_UCG-013 could perhaps even contribute to leanness since the host has to exert a great deal of metabolic effort to regenerate degraded glycoproteins, such as intestinal mucins. Overall, Ruminococcaceae_UCG-013 may be one of the main reasons for the maintenance of lean phenotypes in mice.

## Figures and Tables

**Figure 1 biomedicines-10-03272-f001:**
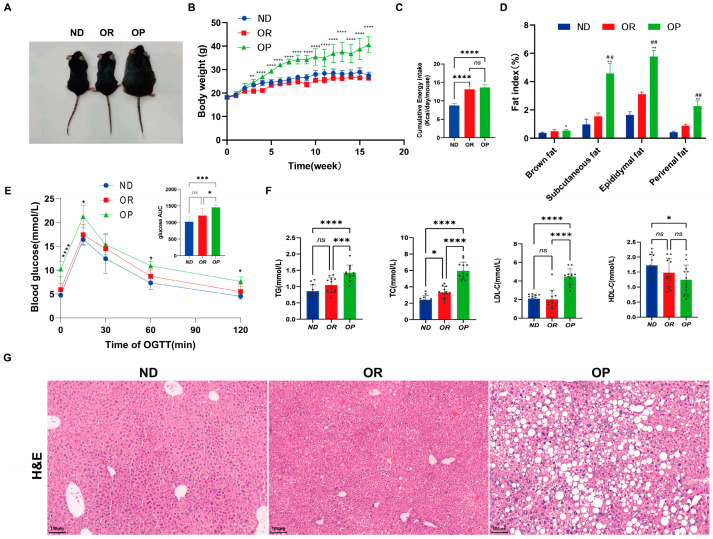
The physiological changes in the Normal diet (ND), obesity-prone (OP), and obesity-resistant (OR) groups. (**A**) Mice representative images. (**B**) Body weight (*n* = 12 per group). ** *p* < 0.01 and **** *p* < 0.0001 vs. the OR group. (**C**) Energy intake. (**D**) Fat index. * *p* < 0.05 and ** *p* < 0.01 vs. the ND group; *##* Fat index was significantly altered in OP relative to OR (*p* < 0.01). (**E**) Blood glucose levels were measured by an oral glucose tolerance test (OGTT) performed during the 16 weeks of the experiment. * *p* < 0.05, *** *p* < 0.001 and **** *p* < 0.0001 vs. the OR group (**F**) Serum biochemical parameters. A sample is represented by each black dot. (**G**) Representative images of H&E staining of liver sections (*n* = 6 per group). The scale bar for H&E staining represents 100 µm. Data are expressed as the mean ± SD. **** *p* < 0.0001, *** *p* < 0.001, ** *p* < 0.01 and * *p* < 0.05 were considered significant. *ns*, not significant.

**Figure 2 biomedicines-10-03272-f002:**
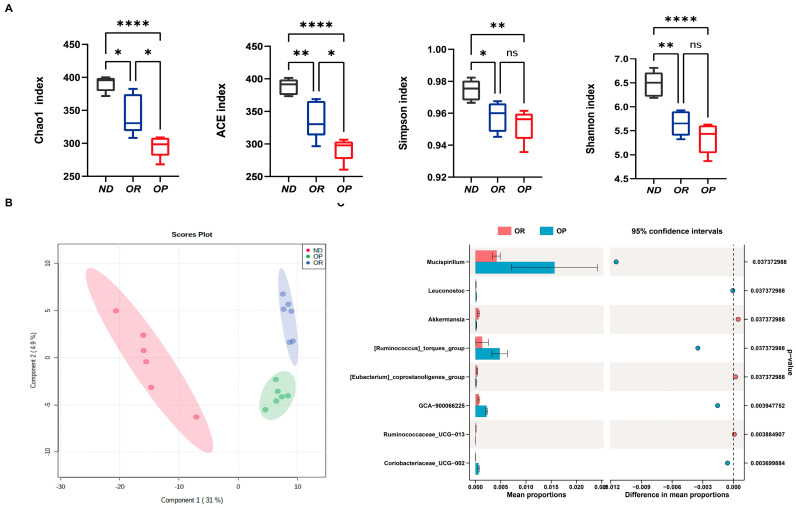
Significantly different microbiota structures were observed between OR and OP groups. (**A**) Comparison of alpha diversity indices in the OR versus OP groups. (**B**) Partial least-square discriminant analysis (PLS-DA) scores plot of OTU level showing the groupings of ND (red) group, OP (green), and HF-OR (blue). (**C**) Extended error bar plot between the OP and OR groups at the genus level. * *p* < 0.05, ** *p* < 0.01 and **** *p* < 0.0001 were considered significant. *ns*, not significant.

**Figure 3 biomedicines-10-03272-f003:**
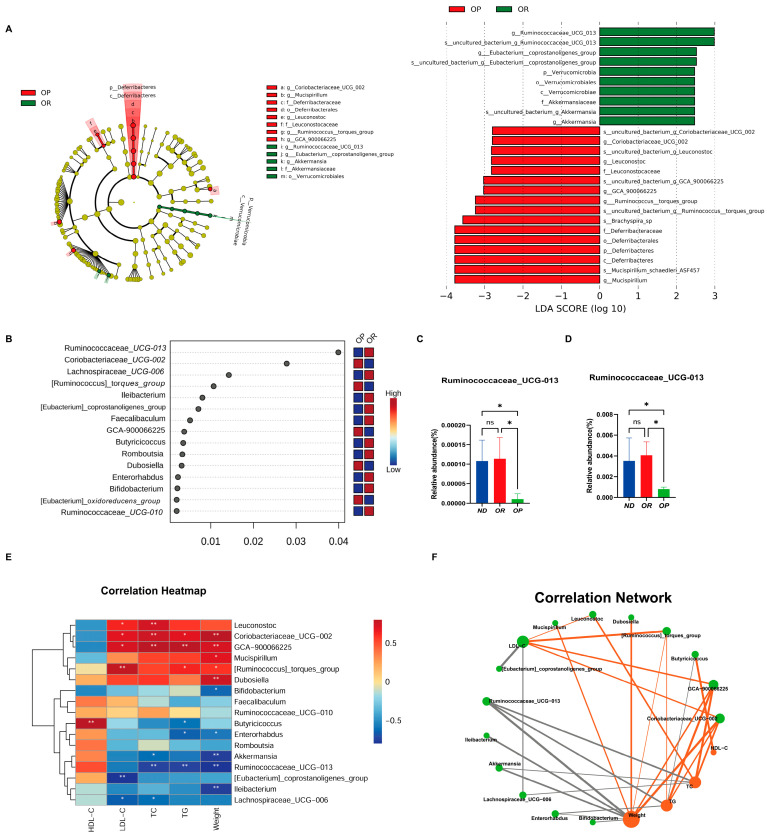
The specific gut microbiota linked to the development of obesity according to the machine learning algorithm. (**A**) Linear discriminant analysis (LDA) effect size (LEfSe) analyses identified bacterial biomarkers of OP and OR mice (LDA > 2, *p* < 0.05). (**B**) Ranking of species importance produced by random forest analyses. (**C**) The relative abundance of Ruminococcaceae_UCG-013 in different groups at week 8. (**D**) The relative abundance of Ruminococcaceae_UCG-013 in different groups at week 16. (**E**) Heatmap of Spearman’s correlation between the faecal microbiota with significant differences and serum biochemical parameters and body weight. An indicator of the degree of association is the intensity of the colour. (**F**) Co-occurrence network. Note: *p* < 0.05, confidence interval = 95%; green nodes are gut microbial genera; orange nodes are serum biochemical indicators; orange and black lines represent positive and negative correlations, respectively. In addition, the width of the lines indicates the strength of the correlation. * *p* < 0.05. ** *p* < 0.01. *ns*, not significant.

**Figure 4 biomedicines-10-03272-f004:**
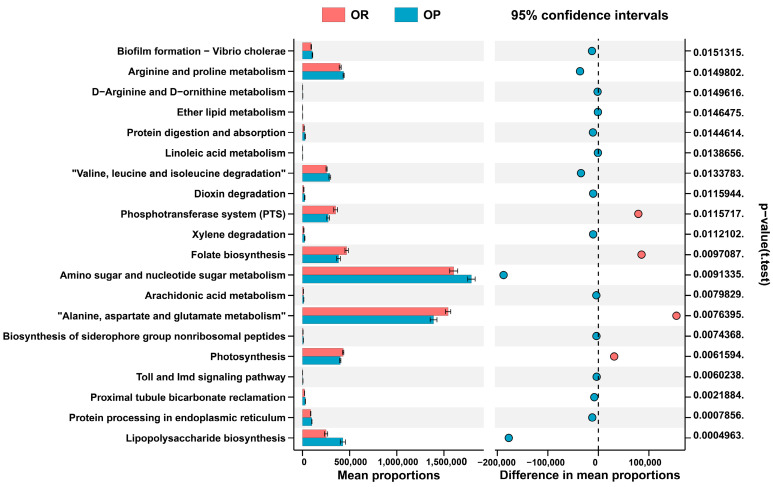
PICRUSt2 analysis predicted metabolic pathways by PICRUSt2 using 16s rRNA gene sequences (Only the first 20 will be displayed). The rightmost is the *p*-value, and *p* < 0.05 is statistically significant.

**Figure 5 biomedicines-10-03272-f005:**
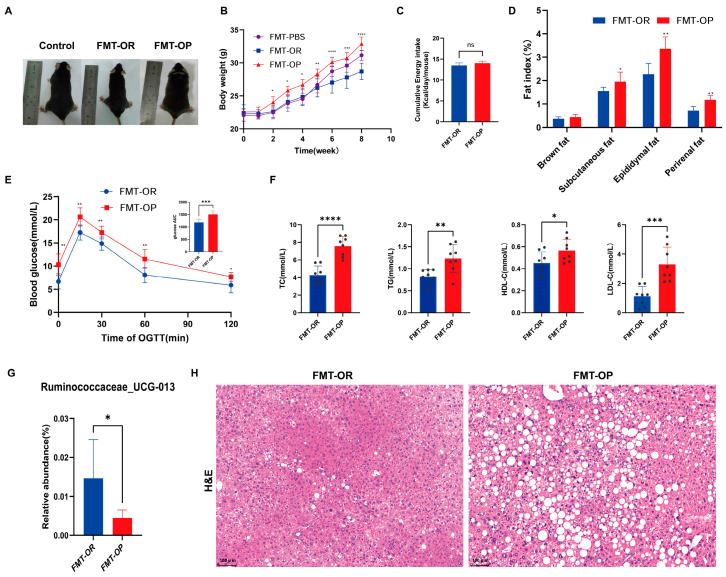
The physiological change and relative abundance of Ruminococcaceae_UCG-013 change in the Mice with obesity-prone fecal flora transplanted (FMT-OP)and Mice with obesity-resistant fecal flora transplanted (FMT-OR) groups. (**A**) Images of mice representative of the species. (**B**) Body weight. * *p* < 0.05, ** *p* < 0.01, *** *p* < 0.001 and **** *p* < 0.0001 vs. the FMT-OR group. (**C**) Energy intake. (**D**) Fat index. (**E**) Blood glucose levels were measured by an oral glucose tolerance test (OGTT) performed in the eight weeks of FMT. (**F**) Serum biochemical parameters. A sample is represented by each black dot. (**G**) The relative abundance of Ruminococcaceae_UCG-013 in different groups at week 8. (**H**) Representative images of H&E staining of liver sections. The scale bar for H&E staining represents 100 µm. Data are expressed as the mean ± SD (*n* = 8 per group). **** *p* < 0.0001, *** *p* < 0.001, ** *p* < 0.01 and * *p* < 0.05 were considered significant. *ns*, not significant.

## Data Availability

16S amplicon data from animal faeces of C57/BL6J mice https://www.ncbi.nlm.nih.gov/bioproject/PRJNA909055 (accessed on 6 December 2022).

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
