# Peer review of "Ruminococcaceae_UCG-013 Promotes Obesity Resistance in Mice"

_biomedicines, 2022, doi:10.3390/biomedicines10123272_

Round 1

Reviewer 1 Report

In the present study, authors investigated effect of gut microbiota on weight gain during HFD feeding in mice.

I think this study have adequately conducted and well written.

Information shown in this manuscript can help to understand.

I have several minor comments.

1.       In Methods. Authors divided HFD group ranked in the upper and lower quartiles for body weight gain. Please inform how to maintain these mice during HFD feeding. How many mice were maintained in a cage?

2.       In Methods. Page 3 line 99, 200-litre is 200-micro-litre?  

3.       In Methods. Page 3 line 115, 80 ºC is -80 ºC?

4.       In Results. Page 4 line 159 to 162. OGTT data from Figure 1D showed that glucose levels were higher in OR group. Does this figure correct?

5.       In Figure 1F. Please show scale bar.

6.       In Figure 1A and D. Please show asterisk in figure if significantly different.

7.       In Results. Page 5 line 188-191. Please show where the data is. Figure 2A?

8.       Figure 3A, B, E, and F. Letters are too small.

9.       In conclusion. The last sentence is an overestimation.

It is not clear that UCG-013 contributes to the maintenance of a lean phenotype in mice.

10.   How to develop the gut microbiota in OR and OP? The genetic background and environment were similar in both groups. What make the difference in the gut microbiota between groups? Please discuss that. This is the point of most interest to the reader.

Finally thank you for your interesting manuscript. I enjoyed reading it.

Author Response

Dear Professor:

On behalf of my co-authors, we would like to thank you very much for giving us the opportunity to revise our manuscript. We greatly appreciate your positive and constructive comments and suggestions on our manuscript entitled "Ruminococcaceae_UCG-013 promotes obesity resistance in mice" (Manuscript ID: Biomedicines-2030434).

We have carefully studied the reviewer comments and made revisions, which are marked in red in the revised manuscript and supplementary material. The point-by-point responses to the reviewers are listed below. We have made every effort to revise the manuscript based on comments and hope for approval. Please refer to the attachment

We greatly appreciate your comments on our documents. Looking forward to hearing from you. Thank you and best regards.

Reviewer 2 Report

The experimental schema seemed correct.

Line 115: minus 80 C

Results are valuable and promising.

Line 299: “In this study, we found that Ruminococcaceae_UCG-013  had a higher relative abundance in the gut microbiota of the lean phenotype than in those  of the obese phenotype. Our findings suggest that Ruminococcaceae_UCG-013 plays an  essential role in obesity resistance.” – I could in 100% agree with the first sentence but the second one is not proven. To do that the obese individuals should be inoculated with Ruminococcaceae_UCG-013 and then the leaning effect must occur.

Please add information if the Ruminococcaceae_UCG-013 are present in human intestine thus those bacteria could be serve as obesity resistance in other species, i.e. humans.

I think that the manuscript could be considered for publication in Biomedicines.

Author Response

Dear Prof: 

On behalf of my co-authors, we thank you very much for giving us an opportunity to revise our manuscript. We appreciate you very much for your positive and constructive comments and suggestions on our manuscript entitled “Ruminococcaceae_UCG-013 promotes obesity resistance in mice(Manuscript ID: biomedicines-2030434).

We have studied reviewer’ comments carefully and have made revisions which were marked in red in the revised Manuscript and Supplementary Material. The point-by-point response to reviewer are listed below. We have tried our best to revise our manuscript according to the comments and hope to meet with approval.

Please refer to the attachment

We would like to express our great appreciation to you for the comments on our paper. Looking forward to hearing from you.

Thank you and best regards.

Yours sincerely,

Reviewer 3 Report

The manuscript submitted by Feng et al., is an interesting in vivo study that investigated the effects of Ruminococcaceae_UCG-013 on obesity in mice.

The manuscript is interesting and reveals potential target and strategies for weight management and potential glycemia control.

The paper is well written and flows well. The reviewer would like to propose some points for the authors' consideration. 

1. Was there a power calculation for the determination of animals?

2. The authors make a mention regarding the glycemic control and the potential implication towards diabetes. The discussion would benefit significantly from a short discussion regarding the microbiome and Type 2 Diabetes Mellitus., the following manuscript could be helpful towards such discussion:

Sikalidis, A.K.; Maykish, A. The Gut Microbiome and Type 2 Diabetes Mellitus: Discussing A Complex Relationship. Biomedicines 2020, 8, 8. https://doi.org/10.3390/biomedicines8010008.

3. Another point that the discussion should consider including is the discussion on immune responses and how those may extend risk towards chronic disease including metabolic regulation. There is also interest in discussing the diet and particularly amino acid intake in this regard. 

Here is another paper to consider: 

  1. Sikalidis AK (2015) Amino Acids and Immune Response: A role for cysteine, glutamine, phenylalanine, tryptophan and arginine in T-cell function and cancer? Pathol Oncol Res21(1):9-17. doi: 10.1007/s12253-014-9860-0.

Author Response

Dear Professor:

On behalf of my co-authors, we would like to thank you very much for giving us the opportunity to revise our manuscript. We greatly appreciate your positive and constructive comments and suggestions on our manuscript entitled "Ruminococcaceae_UCG-013 promotes obesity resistance in mice" (Manuscript ID: biomedicines-2030434).  

We have carefully studied the reviewer comments and made revisions, which are marked in red in the revised manuscript and supplementary material. The point-by-point responses to the reviewers are listed below. We have made every effort to revise the manuscript based on comments and hope for approval.

Please refer to the attachment

We greatly appreciate your comments on our documents. Looking forward to hearing from you. Thank you and best regards. 

Reviewer 4 Report

Well-conducted research, the results of which provide new data on the potential relationship between the composition of the gut microbiota and the risk of obesity. The available data on this subject are inconsistent and further studies are necessary.

Author Response

Dear Prof:

On behalf of my co-authors, we would like to thank you very much for giving us the opportunity to revise our manuscript. We greatly appreciate your positive and constructive comments and suggestions on our manuscript entitled "Ruminococcaceae_UCG-013 promotes obesity resistance in mice" (Manuscript ID: biomedicines-2030434).

We have carefully studied the reviewer comments and made revisions, which are marked in red in the revised manuscript and supplementary material. The point-by-point responses to the reviewers are listed below. We have made every effort to revise the manuscript based on comments and hope for approval.

Please refer to the attachment

We greatly appreciate your comments on our documents. Looking forward to hearing from you.

Thank you and best regards.

Round 2

Reviewer 3 Report

In my professional opinion the authors have done a good job at addressing the reviewer's points and the manuscript has been significantly improved. I thus recommend acceptance for publication.